# Effect of DSS-Induced Ulcerative Colitis and Butyrate on the Cytochrome P450 2A5: Contribution of the Microbiome

**DOI:** 10.3390/ijms231911627

**Published:** 2022-10-01

**Authors:** Stefan Satka, Veronika Frybortova, Iveta Zapletalova, Pavel Anzenbacher, Eva Anzenbacherova, Hana Kozakova, Dagmar Srutkova, Tomas Hudcovic, Lenka Jourova

**Affiliations:** 1Department of Medical Chemistry and Biochemistry, Faculty of Medicine and Dentistry, Palacky University Olomouc, 775 15 Olomouc, Czech Republic; 2Department of Pharmacology, Faculty of Medicine and Dentistry, Palacky University Olomouc, 775 15 Olomouc, Czech Republic; 3Laboratory of Gnotobiology, Institute of Microbiology of the Czech Academy of Sciences, 549 22 Novy Hradek, Czech Republic

**Keywords:** gut–liver axis, gut inflammation, butyrate, hepatic drug metabolism, cytochromes P450, germ-free mice, microbiome

## Abstract

Several studies have indicated the beneficial anti-inflammatory effect of butyrate in inflammatory bowel disease (IBD) therapy implying attempts to increase butyrate production in the gut through orally administered dietary supplementation. Through the gut–liver axis, however, butyrate may reach directly the liver and influence the drug-metabolizing ability of hepatic enzymes, and, indirectly, also the outcome of applied pharmacotherapy. The focus of our study was on the liver microsomal cytochrome P450 (CYP) 2A5, which is a mouse orthologue of human CYP2A6 responsible for metabolism of metronidazole, an antibiotic used to treat IBD. Our findings revealed that specific pathogen-free (SPF) and germ-free (GF) mice with dextran sulfate sodium (DSS)-induced colitis varied markedly in enzyme activity of CYP2A and responded differently to butyrate pre-treatment. A significant decrease (to 50%) of the CYP2A activity was observed in SPF mice with colitis; however, an administration of butyrate prior to DSS reversed this inhibition effect. This phenomenon was not observed in GF mice. The results highlight an important role of gut microbiota in the regulation of CYP2A under inflammatory conditions. Due to the role of CYP2A in metronidazole metabolism, this phenomenon may have an impact on the IBD therapy. Butyrate administration, hence, brings promising therapeutic potential for improving symptoms of gut inflammation; however, possible interactions with drug metabolism need to be further studied.

## 1. Introduction

The inflammatory bowel diseases (IBD) represent chronic and relapsing inflammation of gastrointestinal wall manifesting mainly in two forms, Crohn’s disease and ulcerative colitis (UC), which severely affect patients’ quality of life along with increasing risk of developing cancer [1]. Despite the fact that millions of people suffer from IBD, the pathogenesis and etiology remain poorly understood. Therefore, IBD are classified as multifactorial disorders, the incidence of which is dependent on a combination of genetic, immunological, environmental and microbiological variables. Lines of evidence point to the finding that particularly the microorganisms living in the gut seem to play an essential role in the pathogenesis of UC [2,3]. The dynamic population of microorganisms inhabiting gastrointestinal tract (GIT), referred to as microbiota, and all the genes they carry, called the microbiome, with its complex network of interactions with the host intestinal wall, is being extensively studied for its involvement in human physiology and pathological processes [4]. The largest number and greatest diversity of microbiota with a predominance of anaerobic bacteria is located in the distal part of GIT where also production of the short-chain fatty acids (SCFAs) takes place [5,6].

SCFAs, mainly acetate, propionate and butyrate, are produced during bacterial anaerobic fermentation of dietary fibers and contribute to maintaining intestinal homeostasis and sustaining intestinal barrier integrity, serve as an energy source and may exert anti-inflammatory and anti-carcinogenic properties [7,8,9]. The dysbiotic state of gut microbiota, characterized by imbalance in microbiota diversity and altered composition of fermentation products, is linked to the onset and aggravation of inflammatory conditions including IBD [10].

Accumulating evidence shows that gut microbiota has a certain capacity of drug biotransformation with various consequences on the drug pharmacological activity [11,12,13]. Experiments on germ-free (GF) animals showed that gut microbiota and its metabolites are implicated in the regulation of expression of a host’s hepatic cytochromes P450 (CYPs) [14,15]. CYPs are a superfamily of oxidoreductases taking part in the metabolism of various endogenous and exogenous compounds [16]. Especially, members of CYP1, 2 and 3 families are involved in the biotransformation of xenobiotics and 70–80% clinically used drugs [17]. 

The liver is exposed to all nutrients and microbial metabolites carried by portal blood due to the anatomical interconnection between gut and liver allowing bidirectional communication and signaling [18]. One of the most studied gut-microbiome-derived metabolites is unquestionably butyrate, an endogenous compound produced by bacterial fermentation mainly from dietary fiber in the colon. Considering that butyrate is important for proper function and intestinal health, while liver drug metabolism is sensitive to inflammatory conditions [19], here, we have investigated the impact of microbiota, DSS-induced colitis and the oral application of butyrate on the hepatic drug biotransformation enzyme.

We have focused on the CYP2A5, the mouse orthologue of human CYP2A6, which is responsible for metabolizing metronidazole [20], an antibiotic used to treat IBD in humans [21]. The impact of butyrate administration on the expression and activity of hepatic CYP2A5 in murine liver under physiological and inflammatory conditions was assessed. To determine the contribution of enteric microbiota on CYP2A5 expression and activity in our experimental settings, GF mice lacking the intestinal flora were used in comparison with control specific-pathogen-free (SPF) mice.

## 2. Results

The aim of the in vivo study was to investigate the effect of DSS-induced ulcerative colitis and orally administered sodium butyrate (SB) on the hepatic biotransformation enzyme CYP2A5 in the SPF and GF mice, enabling the assessment of the role of the gut microbiome. Each of the two main groups of animals (SPF and GF) were further subdivided into four other groups according to Figure 1. It was found that the mRNA expression of hepatic CYP2A5 was significantly downregulated by DSS-induced colitis (DSS, group 2) in the SPF mice compared to the control healthy animals (CT, group 1) (Figure 1A). SPF mice pre-treated by two weeks’ administration of SB prior to DSS treatment (SB + DSS, group 3) did not show a different effect of the CYP2A5 expression than DSS mice without SB administration (DSS, group 2); in other words, the CYP2A5 expression remained downregulated compared to the healthy control mice (CT, group 1). The same trend, but non-significant, was observed in GF mice (Figure 1A). SB applied to healthy controls (SB, group 4) did not significantly alter the expression of CYP2A5 in GF mice, but caused a slight decrease in expression in SPF mice (Figure 1C). Furthermore, we studied whether the effect of inflammatory condition and/or microbiome on the CYP2A5 expression and enzyme activity was promoted through nuclear receptor Constitutive androstane receptor (CAR) [22,23]. We showed that DSS-induced colitis significantly downregulates the expression of CAR (DSS, group 2) and SB administered for two weeks at the “preventive” setting (SB + DSS, group 3) did not affect the DSS treatment outcome in both the SPF and GF mice (Figure 1B). The observations mentioned above are in line with the expression pattern of CYP2A5 (Figure 1A). Interestingly, two weeks’ administration of SB to healthy control mice (SB, group 4) caused the significant downregulation of CAR in GF mice (Figure 1D) suggesting that SB affects the expression of CAR only in the absence of microbiota. 

The changes in mRNA expression of CYP2A5 by DSS-induced colitis (DSS, group 2) was reflected in the enzyme activity of CYP2A; however, interestingly, it was different in GF and SPF mice (Figure 2A). The activity of CYP2A was markedly decreased (to 50%) in SPF mice under inflammatory conditions (DSS, group 2), while the administration of SB prior to DSS (SB + DSS, group 3) reversed this effect and activity was recovered almost to the control value (CT, group 1) (Figure 2A). On the other hand, in GF mice, DSS-induced colitis (DSS, group 2) led to a slight increase in CYP2A activity compared to control mice (CT, group 1) and the administration of SB prior to DSS (SB + DSS, group 3) did not change this phenomenon (Figure 2A). In line with mRNA expression, the enzyme activity of CYP2A was not significantly affected by two-week SB administration to healthy controls (SB, group 4) or to SPF or GF mice (Figure 2B). The results indicate that SB administration and inflammatory conditions have a different effect on the hepatic CYP2A in the GF and SPF mice, showing the importance of the presence of microbiota in these processes. 

## 3. Discussion

Clinical and experimental studies indicate that pathogenic alteration of the microbiota composition and function, referred to as dysbiosis, may be a critical factor in the pathogenesis of inflammatory bowel disease (IBD) [24]. Imbalances in microbial diversity are suggested to trigger aberrant immune responses and chronic inflammation in genetically predisposed individuals [2]. Pathological changes in mucosae increases gut permeability allowing products of bacterial metabolism and components from the intestinal wall to enter circulation and accumulate in plasma in higher concentrations. These substances originate from the host’s own cells (damage-associated molecular patterns and pro-inflammatory interleukins) and microbial cells (metabolites and cells’ structural components) mediating various effects on host metabolic and inflammatory pathways in distal tissues including the liver [25,26,27]. 

It has been reported that mice with DSS-induced ulcerative colitis can develop hepatic inflammation [28,29], which may explain the downregulated expression of several hepatic CYPs in such experimental models [30,31]. Also, in GF mice, the IL-1β mRNA expression was slightly but not significantly increased in the liver of mice with DSS-induced colitis (DSS, group 2) compared to control healthy mice (CT, group 1) (Appendix A). The expression of CYP is complex and regulated by several transcription factors including aryl hydrocarbon receptor (AhR), constitutive androstane receptor (CAR), and pregnane X receptor (PXR) [32]. The findings of multiple in vitro and in vivo studies confirm that pro-inflammatory cytokines alter the expression of CYPs and transcription factors involved in their regulation [33,34,35]. In line with these data, we have observed that DSS-induced-colitis-downregulated expression of CYP2A5 (Figure 1A) and CAR (Figure 1C) and even the enzyme activity of CYP2A was decreased in the liver of SPF mice with colitis (Figure 2A). 

Apart from inflammatory conditions, the gut microbiome has become an important factor affecting drug metabolism and bioavailability [12,36,37]. Animal studies have demonstrated that gene expression, protein levels, and activity of drug-metabolizing enzymes are altered in the absence of microbiome [14,15] and could be changed after colonization [38] or by administration of probiotics [39]. The gut microbiome may constitute a promising therapeutic target in the IBD treatment as its involvement in the disease pathology has been proved. From the above mentioned, however, it can be expected, that microbiome-targeted interventions may affect not only microbial composition/function, but also the hepatic drug metabolism. For this reason, GF mice have been used in our experiment to assess both the lack of microbiome (and thus, lack of microbiome-derived metabolites) and inflammatory conditions on the CYP2A5 expression and regulation. At the level of mRNA, we did not see any significant changes in CYP2A5 and CAR expression between GF and SPF mice in both the healthy control group (CT, group 1) and mice with DSS-induced colitis (DSS, group 2) (Figure 1A,B). On the other hand, the enzyme activity of CYP2A in the liver of mice with DSS-induced colitis (DSS, group 2) varied markedly between GF and SPF mice. While a significant decrease in CYP2A activity was observed in SPF mice, in the liver of GF mice, it had a tendency to increase compared to respective healthy control mice (Figure 2A). 

One of the most studied gut-microbiome-derived metabolites is unquestionably butyrate, an endogenous compound produced by bacterial fermentation mainly from dietary fiber in the colon. With respect to its physiological roles, the possible therapeutic effect of butyrate on IBD has been investigated in clinical studies showing that butyrate ameliorates the degree of inflammation in the gut [40,41]. In addition, animal studies presented that preventative administration of butyrate is able to ameliorate inflammation in DSS colitis [28,42]. A common sign of IBD-associated dysbiosis is decreased abundance of butyrate-producing bacteria and decreased levels of SCFAs [43,44]. For this reason, apart from adding fiber to the diet, pharmaceutical dietary supplements based on the oral administration of SB are on the rise as a strategy to enhance butyrate production and to alleviate gut inflammation [45]. Moreover, in vitro and in vivo studies have shown that butyrate is able to affect the expression of many genes through the inhibition of histone deacetylases (HDAC) activity [8]. On top of that, butyrate may interfere with drug metabolism acting on transcription factors involved in the regulation of expression of biotransformation enzymes. It has been recently demonstrated that butyrate affects the gene expression and activity of CYP1A1/2 through the AhR signaling pathway also in primary human hepatocytes [46]. Furthermore, butyrate has been shown to interfere with drug metabolism by influencing hepatic CYP expression and activity in a mice model of DSS-induced colitis [28].

In regard of these findings indicating the effect of butyrate on liver function, we have investigated the effect of butyrate administration to the control healthy mice as well as mice with DSS-induced colitis, in the presence/absence of a microbiome (GF and SPF). In our experiment, SB pre-treatment prior to DSS administration (SB + DSS, group 3) did not show a different effect on the CYP2A5 and CAR mRNA expression compared to the DSS group (DSS, group 2) neither in the GF nor SPF mice (Figure 1A,B). After butyrate administration to healthy control mice, significant downregulation of CAR mRNA expression was observed, but interestingly, only in GF mice (Figure 1D). It has been shown that CAR has an anti-inflammatory function and several endogenous microbial metabolites have an ability to activate it [47]. The fact that butyrate downregulated the expression of CAR only in the absence of the microbiota highlights the important role of bacterial metabolites in CAR regulation. 

Interestingly, the enzyme activity of CYP2A varied markedly between SPF and GF mice, not only by DSS-induced colitis but also by butyrate administration prior to DSS-induced inflammation. In the presence of the microbiome (SPF mice), the administration of butyrate prior to DSS (SB + DSS, group 3) reversed the inhibition effect of DSS and the activity was recovered almost to the control value (CT, group 1) (Figure 2A). Meanwhile, in the GF mice, the DSS-treatment (DSS, group 2) and the combination of butyrate and subsequently DSS (SB + DSS, group 3) caused the same effect and slightly increased the activity compared to the control group (Figure 2A). On the other hand, the administration of butyrate to healthy animals had a negligible effect on the enzyme activity of CYP2A in both SPF and GF mice (Figure 2B). The results indicate that the enzymatic activity of CYP2A is modulated by microbiota under inflammatory conditions while the exact mechanism remains to be elucidated.

The majority of clinically prescribed drugs are predominantly metabolized by hepatic CYP [16]. Since the activity of CYP may be affected by a variety of factors, and due to the anatomical interconnection between the gut and liver, examining the impact of gut inflammation and butyrate supplementation on CYP could be helpful in predicting pharmacokinetics and avoiding undesirable side effects. It has been reported that blood concentration of metronidazole was increased in UC patients [48]. Taking into account that metronidazole is metabolized predominantly by CYP2A6 in humans, elevated levels of metronidazole may imply a decrease in CYP2A6 activity. In our experimental settings (murine model of UC), we showed that activity and gene expression of CYP2A5, the mouse orthologue of human CYP2A6, was significantly reduced by 50%. For that reason, we could expect higher concentrations of metronidazole in murine blood/plasma. Our results may provide a possible explanation for the mechanism underlying the elevated concentration of metronidazole in UC patients [48]. 

Although, mice with DSS-induced colitis are the most widely used and well-described experimental models, sharing many similarities with human UC, the extrapolation of mouse data to humans must be approached with caution. The main limitation of the presented study lies in the methodology. As the gut microbiota seems to be decisive for the inception of UC, inflammation induced by DSS in GF mice may be promoted by a different mechanism than in SPF mice. Under GF conditions, DSS probably does not induce acute or chronic UC, but rather exerts its toxic effect on the intestinal mucosa and impairs barrier function [49,50]. The exact mechanism of DSS’s effect under GF conditions is still unclear. It has to be noted that the different DSS effects in GF and SPF mice may be another factor influencing CYP2A expression and activity. The focus of the study was on the enzymatic activity of cytochrome P450 2A6, as the activity reflects the real “end point” and corresponds better to the potential effect from a clinical point of view. The activity should be expressed in relation to the known concentration of CYPs which makes this approach very demanding on the amount of microsomal fraction. This leads to the necessity to pool the liver samples in each group at the expense of losing information about differences between individuals.

## 4. Materials and Methods

### 4.1. Animals and Experimental Design

Two-month-old female BALB/c mice reared in SPF (specific-pathogen-free) and GF (germ-free) conditions were used. SPF mice were fed by 25 kGy irradiated sterile pellet diet (Altromin, Lage, Germany) and water *ad libitum*. SPF mice were regularly checked for the absence of potential pathogens according to an internationally established standard (FELASA). GF mice were born and housed under sterile conditions in Trexler-type plastic isolators and fed a 50 kGy irradiated sterile pellet diet (Altromin, Lage, Germany) and sterile water *ad libitum*. Axenicity was assessed every two weeks by confirming the absence of bacteria, molds and yeasts by the aerobic and anaerobic cultivation of mouse feces and swabs from the isolators in VL (Viande-Levure), Sabouraud dextrose, and meat peptone broth and subsequent plating on blood, Sabouraud, and VL agar plates. Animals were kept in a room with a 12 h light–dark cycle at 22 °C.

SPF and GF mice were divided into 4 groups (5 mice per group) and treated with sodium butyrate (SB) [51] and/or dextran sulfate sodium (DSS) [49] according to experimental design (Figure 3). A sterile 0.5% SB solution was prepared by dissolving SB (Sigma-Aldrich, Co., St. Louis, USA) in drinking water and filtering through a 0.22 µm filter. The sterile 2.5% DSS solution was obtained by dissolving DSS (M.W. 36–50 kDa; MP Biomedicals, Illkirch, France) in drinking water and autoclaving. The solutions were refreshed every day. Control mice (group 1, CT) were without any treatment. 

All procedures with animals were approved by the Ethics Committee, Ministry of Education of the Czech Republic. Experiments were approved by the Committee for the Protection and Use of Experimental Animals of the Institute of Microbiology, v. v. i., Czech Academy of Sciences of the Czech Republic (approval ID: 21/2018). 

During experiments, the health status of mice and water consumption were monitored every day. The clinical symptoms of colitis were evaluated as: occurrence of diarrhea, rectal prolapses, and rectal bleeding with scoring: 0—no diarrhea or rectal bleeding; 1—pasty stool or mild, no visible rectal bleeding, positive hemocult test; 2—diarrhea or loss of feces, mild visible blood in feces or anus; 3—extensive bleeding from the rectum, rectal prolapses. At the end of experiments, mice were euthanized with isofuranum (IsoFlo, Zoetis, Prague, Czech Republic) followed by cervical dislocation. The colon was removed and its length was measured. The livers were removed, weighed, frozen in liquid nitrogen, and stored at −80 °C until further processing.

Clinical score and histopathology changes in SPF mice have been previously published showing the prophylactic effect of butyrate in a DSS-induced colitis model [28]. In GF mice, DSS administration significantly increased clinical scores and slightly reduced the colon length and preventive SB administration mitigated this effect (Appendix A).

### 4.2. Preparation of Microsomal Fraction

Murine livers were pooled (5 animals per group) into individual groups (according to Figure 3). Microsomal fractions were prepared by differential centrifugation according to the established protocol [52]. Samples of liver tissue homogenate and final microsomal fractions were stored at −80 °C. Concentrations of cytochrome P450 in liver microsomes were determined spectrophotometrically using carbon monoxide [53].

### 4.3. Cytochrome P450 2A Enzyme Activity Assays

The enzyme activity of CYP2A was measured in the mouse hepatic microsomal fractions according to the established method [54]. For the determination of murine CYP2A activity, coumarin was used as a substrate. The incubation mixture contained 100 mM potassium phosphate buffer (pH 7.4), substrate (coumarin) and NADPH-generating system consisting of isocitrate dehydrogenase (0.2 U/mL), NADP^+^ (0.5 mM), isocitrate (4 mM), MgSO_4_ (5 mM) and hepatic microsomal fraction. The amount of microsomal fraction in reaction mixtures corresponded to 35 pmol P450 and the final concentration of coumarin was 10 μM. The concentration of metabolite 7-OH-coumarin was measured using a Shimadzu LC-20 HPLC system (Shimadzu, Kyoto, Japan) with fluorescence detection. The measurements were performed in a LiChrospher RP-18 column (Merck, Darmstadt, Germany). The determinations were carried out in triplicate.

### 4.4. RNA Isolation and Quantitative Real-Time RT-PCR (qRT-PCR)

Total RNA was isolated from murine liver tissues stored in RNA later using an RNeasy Plus Mini Kit (Qiagen, Hilden, Germany) following the manufacturer’s protocol. RNA concentration and purity were quantified spectrophotometrically at 230, 260 and 280 nm using a NanoPhotometer N60 (Implen, Munich, Germany). Altogether, 1 µg of total RNA was converted to cDNA using a Transcriptor High Fidelity cDNA synthesis kit (Roche, Prague, Czech Republic) according to the manufacturer’s instrumentations. The qPCR was performed in a LightCycler 1536 Instrument (Roche, Prague, Czech Republic) using specific TaqMan Gene Expression Assays (primers: Mm01283978_m1, Mm03024075_m1, Mm00487248_g1, Mm00434228_m1) (Applied Biosystems, Prague, Czech Republic). The 1536-well plates were pipetted using Echo Liquid Handler (Labcyte, Dublin, Ireland). The gene expression of CYP2A5, CAR and IL-1β was normalized to expression of the housekeeping gene—hypoxanthine guanine phosphoribosyl transferase (Hprt). Expressions of target genes were calculated using the ΔΔCT method [55] as a fold change in the treatment groups relative to the control.

### 4.5. Statistical Analysis

The normal distribution of data was tested using the Shapiro–Wilk test. For statistical evaluation, a one-way ANOVA followed by Tukey’s post hoc test was conducted using the Statistica 12 software (Dell, obtained from StatSoft, Prague, Czech Republic). Data are expressed as means ± SD. Differences were regarded as statistically significant when the *p* value was lower than 0.05. Due to the scarcity of material, the statistical significance of activity assays could not be determined.

## 5. Conclusions

Our data indicate that the gut microbiome and its metabolites play an important role in the regulation of CYP2A enzyme under inflammatory conditions which may have possible clinical relevance for metronidazole biotransformation in humans. Further studies are needed to better understand the influence of the microbiome-targeted interventions on the hepatic drug metabolism under the inflammatory state common for complex disorders such as IBD.

## Figures and Tables

**Figure 1 ijms-23-11627-f001:**
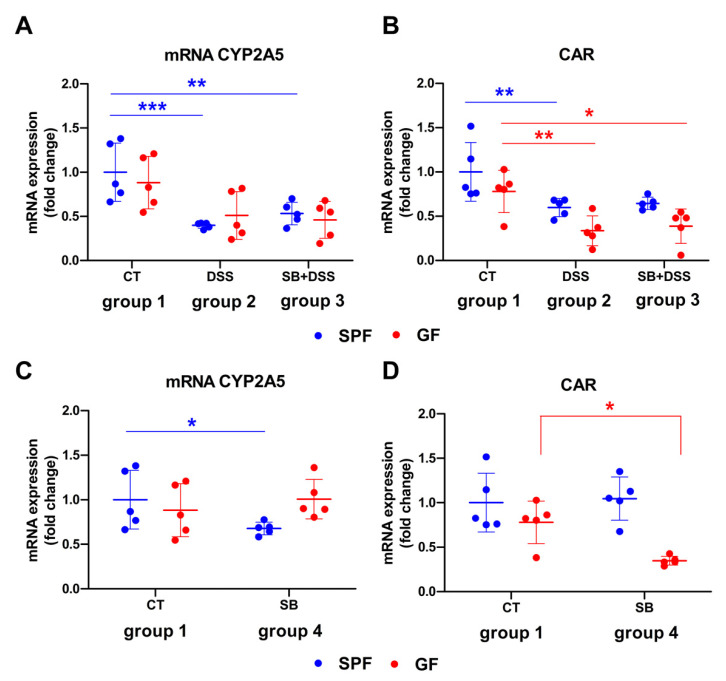
Comparison of mRNA expression of CYP2A5 and CAR in murine liver of GF and SPF mice. Gene expression of (**A**,**C**) CYP2A5 and (**B**,**D**) constitutive androstane receptor (CAR). Results are expressed relative to SPF healthy control group (CT) and represent means ± SD. Significance of differences from the control was determined using one-way analysis of variance (ANOVA) with the Tukey’s multiple post-hoc test being used for comparison of experimental groups to respective SPF and GF controls (*** *p* < 0.001, ** *p* < 0.01, * *p* < 0.05). CT (group 1): drinking water for 1 week; DSS (group 2): 1 week 2.5% DSS in drinking water; SB + DSS (group 3): 0.5% SB in drinking water for 2 weeks followed by a week of 2.5% DSS in drinking water; SB (group 4): 0.5% SB in drinking water for 2 weeks.

**Figure 2 ijms-23-11627-f002:**
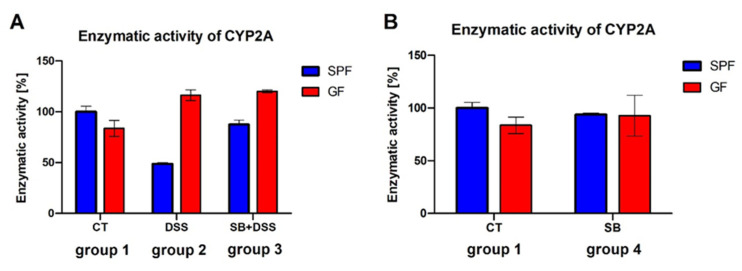
Enzyme activity of CYP2A in the murine liver of SPF and GF mice. Comparison of CYP2A activity of control mice to (**A**) experimental groups with DSS-induced colitis and (**B**) experimental groups treated by SB. Samples were measured in triplicate in a pooled liver microsomal fraction from 5 animals using an HPLC system with UV or fluorescence detection. Data represent the mean ± SD. CT (group 1): drinking water for 1 week; DSS (group 2): 1 week 2.5% DSS in drinking water; SB + DSS (group 3): 0.5% SB in drinking water for 2 weeks followed by a week of 2.5% DSS in drinking water; SB (group 4): 0.5% SB in drinking water for 2 weeks.

**Figure 3 ijms-23-11627-f003:**
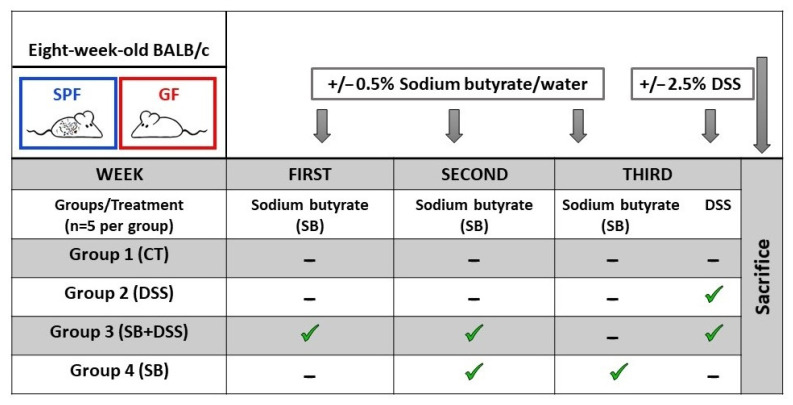
Experimental design. Both SPF (n = 20) and GF (n = 20) mice were subdivided into 4 equal groups. Healthy control mice (group 1, CT) were without any treatment. DSS mice (group 2, DSS) were without SB treatment only with induced colitis by one-week DSS administration. In the third group, SB was administered for two weeks at the “preventive” setting and subsequently, colitis was induced by one week of 2.5% DSS administration (group 3, SB + DSS). In the last group, mice received SB solution for two weeks (group 4, SB) before sacrifice.

## Data Availability

The raw data supporting the conclusions of this manuscript will be made available by the authors, without reservation, to any qualified researcher.

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
