# Peer review of "Effect of DSS-Induced Ulcerative Colitis and Butyrate on the Cytochrome P450 2A5: Contribution of the Microbiome"

_ijms, 2022, doi:10.3390/ijms231911627_

Round 1

Reviewer 1 Report

Interesting original paper presenting a new experimental approach to clarify clinical aspects of the treatment (metronidazole concentration) in IBD patients.  

I have two comments:

1. Authors should check the use of abbreviations in the text, e.g. line 39 UC - abbreviation not introduced before, GF abbreviation introduced line 72, but earlier the word germ-free (line 54) appeared, SB - abbreviation introduced in line 82 without description, then was the word sodium butyrate used without an abbreviation (line 132, 176), abbreviation for CYP2A5 or CYP2a5?, one should make up his mind and use one abbreviation throughout the text 2. The authors should include a subsection showing the limitations of the study, which should focus on the size of the study and statistical analysis - statistical analysis was carried out on a few samples (n = 5), in the case of CYP2A concentration or enzymatic activity, the liver samples were pooled (basically only one sample was available). Is it possible to talk about reliable statistics with n = 5 and assess the normal distribution in such a small sample? For this reason, authors should describe some doubts in the interpretation of the results.

Author Response

Reviewer 1

Interesting original paper presenting a new experimental approach to clarify clinical aspects of the treatment (metronidazole concentration) in IBD patients.  

We wish to thank the reviewer for appreciation of our manuscript.

I have two comments:

  1. Authors should check the use of abbreviations in the text, e.g. line 39 UC - abbreviation not introduced before, GF abbreviation introduced line 72, but earlier the word germ-free (line 54) appeared, SB - abbreviation introduced in line 82 without description, then was the word sodium butyrate used without an abbreviation (line 132, 176), abbreviation for CYP2A5 or CYP2a5?, one should make up his mind and use one abbreviation throughout the text

We apologize for the mistake and thank the reviewer for the note. We have rewritten the abbreviations (highlighted in yellow in Revised Manuscript).

  1. The authors should include a subsection showing the limitations of the study, which should focus on the size of the study and statistical analysis - statistical analysis was carried out on a few samples (n = 5), in the case of CYP2A concentration or enzymatic activity, the liver samples were pooled (basically only one sample was available). Is it possible to talk about reliable statistics with n = 5 and assess the normal distribution in such a small sample? For this reason, authors should describe some doubts in the interpretation of the results.

This is a valuable comment. We added a short paragraph to the Discussion section to consider limitations of the study (lines 225-240).

5 mice per group has been used and the Shapiro-Wilk test has proved normal distribution of our data. However, the reviewer is correct, that due to small sample size and other limitations our conclusion should be presented carefully. On the other hand, we feel that the study design and the results support sufficiently conclusions and suggestions we made. The DSS model of ulcerative colitis is very well characterized, moreover, clinical score and other parameters have shown the induction of inflammation (we added this information to the manuscript, Lines 265-277). For this reasons and due to the high demands of maintaining GF mice we have used 5 mice per groups (altogether about 50 mice for the study). 

Another limitation of this study is about the methodology of CYP2A enzyme activity determination. This approach, to be correct and reliable, is itself very demanding on the amount of microsomal fraction. Considering that mice have small livers we have to make a pooled liver sample to be able to obtain enough volume of microsomal fraction that would allow us to assess the P450 concentration and enzyme activity. Unfortunately, this had to be done at the expense of losing information about differences between individuals. Therefore, the standard deviation in graph showing activities (Figure 2AB) are actually depicting “Measurement precision” – repeated analysis of a single sample not dispersion of individual values. This is explained in Materials and Methods – the statistical analysis and figures on the CYP activities show only a statistical significance of activity assay (Lines 325-326). Although we could not perform appropriate statistical analysis showing biological variation, we believe that the changes in enzyme activity of CYP2A between groups may be significant from drugs pharmacokinetics point of view (Fig. 2A).

We thank again the reviewer for the helpful comments and for taking the time to point out options to improve our manuscript.

Reviewer 2 Report

In this work the authors evaluated the impact of DSS-induced ulcerative colitis and butyrate on the activity of the hepatic biotransformation enzyme CYP2A5, and how this effect is modulated by gut microbiome. By using both SPF and GF mice the authors concluded that gut microbiome is indispensable for the butyrate-mediated reversion of CYP2A activity, that is decreased after DSS administration. This is relevant in the context of IBD pathology since CYP2A5 is the mouse orthologue of the human CYP2A6, which is responsible for the hydroxylation of metronidazole, an antibiotic used to treat IBD in humans. As butyrate arises as a therapeutic approach in IBD, due to its anti-inflammatory properties, understanding the impact of this treatment on the pharmacokinetics and biotransformation of metronidazole, with consequences on the therapy outcome, is of utmost importance.

Major comments:

- Throughout the manuscript the DSS administration is referred to as the inflammatory condition. As this is the case in SPF animals, where this model is very well characterized in terms of phenotypic and molecular changes, DSS induces colon injury and a strong inflammatory response. In GF mice DSS-induced colonic inflammation and overall immune response are greatly diminished (https://academic.oup.com/ecco-jcc/article/10/11/1324/2480001; https://www.nature.com/articles/s41564-022-01094-z#Sec29) . Therefore, the authors should characterize and describe the induction of colitis by DSS in both SPF and GF mice, particularly: body weight variations, histopathological changes in colonic mucosa, and the intestinal and hepatic expression of inflammatory markers. This is essential to understand and correlate the changes observed in Cyp2a5 expression and activity with disease status, when comparing SPF and GF responses to the different stimuli. Results and discussion should be revised considering this perspective.

- Several transcription factors have been described to regulate Cyp2a5 expression. What was the rationale to only evaluate CAR? Given that, at least for the butyrate condition, no correlation between CAR and Cyp2a5 mRNA levels was observed, it seems that DSS and butyrate are affecting Cyp4a5 mRNA levels by distinct mechanisms. Besides CAR mRNA levels, have the authors evaluated other known regulators, as mentioned in the “Discussion”, such as PXR, AhR or even Nrf2? The latter one is particularly interesting due to the connection between the Nrf2/antioxidant response element system and the expression of inflammatory mediators.

- In the experimental design both DSS and butyrate were administered in the drinking water. Were there any differences in water intake between the groups? This should be clarified and described in the Materials and Methods section as well as in the results section.

- Only female mice were used. Does Cyp2a5 presents sexual dimorphism? The rationale is not clear.

Minor comments:

- The Introduction is well structured with adequate references. As the field of gut microbiome interactions is rapidly growing it is important to clarify the meaning of the different terms (https://microbiomejournal.biomedcentral.com/articles/10.1186/s40168-020-00875-0) . The term “microbiome” not only refers to the microorganisms involved (microbiota) but also to the whole spectrum of molecules produced by the microorganisms, including their structural elements (nucleic acids, proteins, lipids, polysaccharides), metabolites, etc. This should be corrected in the introduction.

- In section 4.4. the reference of the different TaqMan probes should be listed.

- In line 281 “mRNA” should be replaced by “cDNA”.

Author Response

Reviewer 2

In this work the authors evaluated the impact of DSS-induced ulcerative colitis and butyrate on the activity of the hepatic biotransformation enzyme CYP2A5, and how this effect is modulated by gut microbiome. By using both SPF and GF mice the authors concluded that gut microbiome is indispensable for the butyrate-mediated reversion of CYP2A activity that is decreased after DSS administration. This is relevant in the context of IBD pathology since CYP2A5 is the mouse orthologue of the human CYP2A6, which is responsible for the hydroxylation of metronidazole, an antibiotic used to treat IBD in humans. As butyrate arises as a therapeutic approach in IBD, due to its anti-inflammatory properties, understanding the impact of this treatment on the pharmacokinetics and biotransformation of metronidazole, with consequences on the therapy outcome, is of utmost importance.

Major comments:

Throughout the manuscript the DSS administration is referred to as the inflammatory condition. As this is the case in SPF animals, where this model is very well characterized in terms of phenotypic and molecular changes, DSS induces colon injury and a strong inflammatory response. In GF mice DSS-induced colonic inflammation and overall immune response are greatly diminished (https://academic.oup.com/ecco-jcc/article/10/11/1324/2480001; https://www.nature.com/articles/s41564-022-01094-z#Sec29).

Therefore, the authors should characterize and describe the induction of colitis by DSS in both SPF and GF mice, particularly: body weight variations, histopathological changes in colonic mucosa, and the intestinal and hepatic expression of inflammatory markers.  This is essential to understand and correlate the changes observed in Cyp2a5 expression and activity with disease status, when comparing SPF and GF responses to the different stimuli. Results and discussion should be revised considering this perspective.

The authors are grateful the reviewer for pointing out at this question. We have added the information about clinical score evaluation and interleukin-1beta mRNA expression to the Methods section (Lines 265-277). We have observed increase in clinical score in DSS and DSS+SB groups compared to control group in GF mice. But, we agree with the reviewer that the DSS-induced colitis under GF condition may be complicated. We have mentioned this phenomenon in the Discussion describing the limitations of the study (Lines 225-240). It seems that enteric microbiota is decisive for the inception of UC. Under GF conditions DSS probably does not induce acute or chronic UC, but rather exerts its toxic effect on the intestinal mucosa and impairs barrier function. The exact mechanism of DSS effect under GF condition is still unclear. We agree with the reviewer and understand his/her point that the different mechanism of DSS effect in GF and SPF mice may as well contribute to the effect on CYP2A expression and activity. Further studies are needed to better understand the complex mutual relationship between modulation of gut microbiota under inflammatory conditions and hepatic drug metabolism. These studies may provide valuable answers with implication for clinical practice.

The histopathological changes in colonic mucosa in SPF after DSS and/or SB administration have been previously published (https://pubmed.ncbi.nlm.nih.gov/35928257/) showing prophylactic effect of butyrate. We performed the analysis with the GF mice as well, however, in this paper we focused on the effect of inflammation and/or SB on the hepatic drug metabolism (more precisely on the CYP2A5 expression and enzyme activity). We believe that adding the histopathological changes evaluation to the Methods Section and discussing the effects on the gut mucosa is beyond the scope of this paper.

Several transcription factors have been described to regulate Cyp2a5 expression. What was the rationale to only evaluate CAR? Given that, at least for the butyrate condition, no correlation between CAR and Cyp2a5 mRNA levels was observed, it seems that DSS and butyrate are affecting Cyp2a5 mRNA levels by distinct mechanisms. Besides CAR mRNA levels, have the authors evaluated other known regulators, as mentioned in the “Discussion”, such as PXR, AhR or even Nrf2? The latter one is particularly interesting due to the connection between the Nrf2/antioxidant response element system and the expression of inflammatory mediators.

We evaluated mRNA expression of CAR for its prominent involvement in the regulation of Cyp2A5 expression (https://pubmed.ncbi.nlm.nih.gov/34880749/) and for its sensitivity to inflammatory conditions [ref. 30]. The results of our study are in agreement with those of Kusunoki et al., 2014 [ref. 31] showing similar patterns in CAR expression in livers of DSS mice, and the same was true for mRNA expression of CYP2A5 in our study. The downregulation was observed for the both mRNA expression of CAR and CYP2A5 also in the group SB+DSS, in other words, SB pretreatment had no effect here. It is true that when butyrate is applied to the healthy mice (without DSS) the mRNA expression pattern is different for CYP2A5 and CAR depending of presence/absence of microbiota (SPF/GF) mice. It may be interesting point for the potential clinical usage of butyrate that its effect on expression and enzyme activity on CYP2A6 (which metabolizes metronidazole) may be significantly different under inflammatory and healthy conditions.

The reviewer is correct. Also other transcription factors may be involved in CYP2A5 expression including PXR, Nrf2, and AhR. However, we did not measure these in our study. The exact mechanism of CYP2A5 regulation is not fully understood yet. Due to so many factors playing role in the inflammatory processes and microbial metabolites action in gut-liver axis it is at current state of knowledge hardly possible to give more precise explanation. We wanted to avoid unnecessary speculation about mechanisms that have not yet been sufficiently confirmed. Also due to the scope of Brief Report we cannot further explore and discuss the molecular mechanism.

In the experimental design both DSS and butyrate were administered in the drinking water. Were there any differences in water intake between the groups? This should be clarified and described in the Materials and Methods section as well as in the results section.

During the whole experiment, we have monitored water consumption every day and any significant differences were observed among groups.

Only female mice were used. Does Cyp2a5 presents sexual dimorphism? The rationale is not clear.

Some evidences of sex-based differences in CYP2A6 expression and enzyme activity can be found in the literature. For example, it has been found that women metabolize nicotine (substrate of CYP2A6) faster than men, hypothesizing that hepatic CYP2A6 activity is induced by female sex hormones (https://pubmed.ncbi.nlm.nih.gov/16678549/). In another study using human livers, female livers had significantly higher CYP2A6 protein and mRNA expression compared to males (https://pubmed.ncbi.nlm.nih.gov/20012030/).

Most of studies used male animal models, however, IBD affects men and women equally (even perhaps with a slight predominance in women) (https://pubmed.ncbi.nlm.nih.gov/31995797/), therefore we used female mice for our study.

Minor comments:

The Introduction is well structured with adequate references. As the field of gut microbiome interactions is rapidly growing it is important to clarify the meaning of the different terms (https://microbiomejournal.biomedcentral.com/articles/10.1186/s40168-020-00875-0). The term “microbiome” not only refers to the microorganisms involved (microbiota) but also to the whole spectrum of molecules produced by the microorganisms, including their structural elements (nucleic acids, proteins, lipids, polysaccharides), metabolites, etc. This should be corrected in the introduction.

Thank you for the note. It is true that the definition is more complex and the sentence where microbiome/microbiota are used as synonyms (lines 38-41) may be confounding and we have rewritten it. We are aware that microflora, microbiota and microbiome are defined differently sometimes (although in the definitions is not complete consensus).

The mentioned article proposed to use the term microbiome for all activity of microorganisms including metabolites produced by them. However, we hesitate to use it in our paper as butyrate under this definition should be refer as “microbiome” as well.

In our article, we compare the effects in the presence or absence of microorganisms (GF/SPF) mice; we do not focus in detail on the exact molecular mechanism and we cannot make clear conclusion which microorganism/bacterial component/metabolite is responsible for the effects. For this reason, we feel that the terms microflora/microbiota/microbiome can be mutually replaceable for presented study.

In section 4.4. the reference of the different TaqMan probes should be listed.

We have incorporated the references to the Method section (Line 313-314).

In line 281 “mRNA” should be replaced by “cDNA”.

Thank the reviewer for the note, we have corrected the mistake.

Thank you for your kind revision and suggestions for improvement of our paper. We appreciate all your time given to our article. As we mentioned above, we concur with you and we will further study the molecular mechanisms behind the effects of gut microbiome on the drug metabolism under physiological as well as inflammatory conditions.

Round 2

Reviewer 2 Report

In line 235 "cytochrome P450 2A6" should be replaced by "cytochrome P450 2A5"

Author Response

In line 235 "cytochrome P450 2A6" should be replaced by "cytochrome P450 2A5"

Thank the reviewer for the note, we have corrected the mistake.  

The english has now been edited.   We thank again the reviewer for the helpful comments and for taking the time to point out options to improve our manuscript.